# sFlt-1, Not PlGF, Is Related to Twin Gestation Choronicity in the First and Third Trimesters of Pregnancy

**DOI:** 10.3390/diagnostics11071181

**Published:** 2021-06-29

**Authors:** Szymon Kozłowski, Anna Stelmaszczyk-Emmel, Iwona Szymusik, Aleksandra Saletra-Bielińska, Robert Brawura-Biskupski-Samaha, Paweł Pietruski, Agnieszka Osińska, Katarzyna Kosińska-Kaczyńska

**Affiliations:** 1University Center for Woman and Newborn Health of the Medical University of Warsaw, Starynkiewicza Square 1/3, 02-015 Warsaw, Poland; Kozlowski1979@go2.pl; 2Department of Obstetrics and Gynecology, Institute of Mother and Child, Kasprzaka 17a, 01-211 Warsaw, Poland; 3Department of Laboratory Diagnostics and Clinical Immunology of Developmental Age, Medical University of Warsaw, Żwirki i Wigury 63A, 02-091 Warsaw, Poland; anna.stelmaszczyk-emmel@wum.edu.pl; 41st Department of Obstetrics and Gynecology, Medical University of Warsaw, Starynkiewicza Square 1/3, 02-015 Warsaw, Poland; iwona.szymusik@wum.edu.pl (I.S.); aleksandra.saletra@gmail.com (A.S.-B.); robertsamaha@gmail.com (R.B.-B.-S.); 5Independent Public Complex of Healthcare Institutions in Pułtusk, 3 Maja 5, 06-100 Pułtusk, Poland; ppietruski@gmail.com; 62nd Department of Obstetrics and Gynecology, Center of Postgraduate Medical Education, Cegłowska 80, 01-809 Warsaw, Poland; a_osinska@interia.pl

**Keywords:** twin pregnancy, PlGF, sFlt-1, sEng, preeclampsia, chorionicity

## Abstract

Background: Preeclampsia occurs more often in dichorionic than in monochorionic twin pregnancy. We hypothesize that serum concentrations of biomarkers: placental growth factor (PlGF), serum soluble fms-like tyrosine kinase-1 (sFlt-1), and endoglin (Eng) differ between monochorionic and dichorionic twin pregnancies. Methods: A prospective observational study including 43 monochorionic and 36 dichorionic twin gestation was conducted. Blood samples were collected twice from all participants: between 11 + 0 and 13 + 6 and between 32 + 0 and 34 + 0 weeks of gestation. PlGF, sFlt-1 and Eng were measured using immnunoenzymatic assays. Results: We found a significantly higher concentration of sFlt-1 in dichorionic in comparison to monochorionic pregnancies in both the first and third trimesters. PlGF and sEng levels did not differ between mono- and dichorionic gestation in both study periods. sFlt-1 level was related to twin gestation chorionicity, while PlGF expression was not. PlGF, sFlt-1 and sEng concentrations increased significantly during gestation and were much higher in the third trimester compared to the values measured in the first trimester. Conclusions: Angiogenic biomarkers expression differ between dichorionic and monochorionic twin pregnancy. The sFlt-1 level is related to chorionicity of a twin gestation.

## 1. Introduction

Preeclampsia (PE) is one of the major causes of maternal and fetal morbidity and mortality worldwide [1,2]. Abnormal placentation is a key hypothesis of PE pathogenesis [3]. Placental dysfunction results in al altered expression of placental proteins, which lead to generalised maternal endothelial injury [3]. A few of these proteins were proved to play a role in PE development in singleton gestation: placental growth factor (PlGF), serum soluble fms-like tyrosine kinase-1 (sFlt-1) and endoglin (Eng) and, therefore, they serve as biomarkers of PE [4]. PlGF is one of the growth factors from the vascular endothelial growth factor (VEGF) family, which promotes vessel formation and development. It is produced by villous cytotrophoblastic tissue and the syncytiotrophoblast [5]. PlGF serum levels increase during singleton pregnancy, with highest expression at about 30–32 weeks of gestation, followed by a decrease [3]. sFlt-1 acts as a soluble receptor for VEGF and therefore decreases the concentrations of circulating VEGF and PlGF [6]. During a normal singleton pregnancy, sFlt-1 level is maintained at a plateau of up to 32 weeks and then increases [6]. Eng is a transmembrane glycoprotein, which plays a role as an accessory receptor for the transforming growth factor-beta (TGF-β). It affects the signalling pathways of TGF-β and endothelial nitric oxide synthase and, therefore, have a significant influence on the angiogenic processes [7].

PlGF and sFlt-1 are successfully used in common clinical practice. In the PROGNOSIS study published in 2016, sFlt-1:PlGF ratio <38 was proposed as short-term prediction of the absence of PE in women with a singleton pregnancy in whom the syndrome is clinically suspected [8].

Multiple gestation is a risk factor of developing PE [9,10,11,12]. Laine et al. published a large study on 16,174 twin pregnancies in which the risk of PE in twins was found to be almost four times higher in comparison with singletons, even after adjustment for other risk factors [13]. Several studies have shown that PE is more common in a dichorionic twin pregnancy in comparison to a monochorionic one [11,14,15,16,17]. On the basis of such an observed relationship, we hypothesize that serum concentrations of PlGF, sFlt-1, sFlt-1/PlGF index and soluble Eng (sEng) differ between monochorionic and dichorionic twin pregnancies.

## 2. Materials and Methods

A prospective observational study was conducted at the University Health Centre of the Woman and Newborn in Warsaw, Poland, between October 2016 and December 2018 and at the 2nd Department of Obstetrics and Gynecology, Center of Postgraduate Medical Education, Warsaw, Poland, between March and December 2020. Women in twin gestation, who gave an inform written consent to participate in the study, were followed from the first trimester of pregnancy to delivery. Gestational age was calculated on the basis of a first day of last menstrual period or a transfer day in assisted reproduction techniques procedures and verified by the crown-rump length (CRL) measured on the first trimester scan (if estimated due dates were inconsistent and the difference was bigger than 5 days, the ultrasound measurement was of primary importance; in the case of CRL discordance, the measurement from the larger twin was chosen). All women were counselled in the outpatient clinic at the University Health Centre of the Woman and Newborn and at the 2nd Department of Obstetrics and Gynecology, Center of Postgraduate Medical Education, those in dichorionic pregnancies routinely once every 4 weeks and in monochorionic pregnancies once every 2 weeks, including an ultrasound scan.

Blood samples were collected twice from all participants. The first collection was made between 11 + 0 and 13 + 6 weeks of gestation and the second one between 32 + 0 and 34 + 0 weeks of gestation. We collected 10 mL samples of venous blood in polystyrene tubes and mixed it with tripotassium versenate (K3-EDTA). Blood samples were centrifuged for 10 min at 1500× *g* at 4 °C within 30 min from collection. The plasma was then frozen at −80 °C. After completing the study group in defrosted plasma PlGF, sFlt-1 and sEng were measured using an immnunoenzymatic assays. PlGF concentration was analysed using an enzyme-linked immunosorbent assay (ELISA) Kit for Placenta Growth Factor (Cloud-Clone Corporation, Katy, TX, USA). The lower limit of sensitivity was 15.6 pg/mL. sFlt-1 level was measured with an ELISA Kit for Vascular Endothelial Growth Factor Receptor 1 (Cloud-Clone Corporation, Katy, USA) with lower limit of sensitivity at 78 pg/mL. sEng concentration was assessed with ELISA Kit for Endoglin (Cloud-Clone Corporation, Katy, USA). The lower limit of measurement was 0.156 ng/mL. The assays were performed following the manufacturer’s instructions. In all the kits, intra-assay precision was <5% and inter-assay precision <10%.

The inclusion criteria contained: age over 18 years old, twin pregnancy, chorionicity established and documented on the 1st trimester sonographic scan (two gestational sacs or lambda sign for dichorionic pregnancy; single gestational sac or T sign for monochorionic pregnancy), verified gestational age, delivery beyond 32 weeks of gestation and complete medical data on the pregnancy outcome. Pregnancies complicated by one or two fetal demises, miscarriage, delivery prior to 32 weeks of gestation, twin to twin transfusion syndrome (TTTS), twin anemia-polycythemia sequence (TAPS), twin reversed arterial perfusion syndrome (TRAP) as well as monochorionic monoamniotic ones were excluded from the study. Only women from whom two blood samples in the first and the third trimester of pregnancy were collected were included in the final analysis. In cases of patients, who delivered at University Health Centre of the Woman and Newborn or at the 2nd Department of Obstetrics and Gynecology, Center of Postgraduate Medical Education, both outpatient and hospital records were reviewed in order to gain complete medical data. If a woman gave birth at any other hospital, a telephone interview was conducted after delivery to obtain all important information.

Smoking status was declared by participants. Past smoking habit was defined as quitting smoking at least one month before the current pregnancy. Discordant twin growth was defined as twin birthweight difference exceeding 25% of the larger twin in each twin pair. PE was diagnosed according to American College of Obstetricians and Gynecologists guidelines [18], whereas gestational diabetes mellitus (GDM), according to the Polish Society of Obstetricians and Gynecologists recommendations [19]. Body mass index (BMI) was calculated by dividing the body mass by the square of the body height. Obesity was defined as BMI 30 and greater. Gestational weight gain was defined as a difference between weight at delivery and pre-gravid weight. Pre-gravid weight was self-reported, while weight at delivery was measured by the hospital stuff. sFlt-1/PlGF ratio was calculated by divided sFlt-1 value by PlGF value.

The study protocol was approved by the Ethics Committees at the Medical University of Warsaw and at the Center for Postgraduate Medical Education and was conducted according to the Declaration of Helsinki.

A power analysis was performed to assess the group size. For the 80% power with error probability 0.05 basing on the previously publish results sample size of 74 (37 in each group) was adequate to obtain statistical significance of the study results (G*Power version 3.1.9.7). Variables are described as mean (±SD), median or percentage, median (interquartile range). The Mann–Whitney test and the Fisher’s exact test were used for the statistical analysis. Box-and-whisker plots were created to visualize results. *p*-values < 0.05 were considered significant. Data were analyzed using Statistica version 13.1.

## 3. Results

The University Health Centre of the Woman and Newborn cohort included 100 women; 56 in monochorionic and 44 in dichorionic twin pregnancy. At the 2nd Department of Obstetrics and Gynecology, Center of Postgraduate Medical Education, 12 women in dichorionic twin pregnancy were recruited to the study. Jointly, 112 women were included; 2 women miscarried, 11 delivered prior to 32 weeks of gestation; 6 were lost in follow up.;14 women were excluded due to TTTS or TAPS diagnosis.

Finally complete data were available in 79 pregnant women and those individuals were included in the study; 43 were in monochorionic (all from the University Health Centre of Woman and Newborn cohort) and 36 in dichorionic gestation (29 from the University Health Centre of the Woman and Newborn cohort and 7 from the 2nd Department of Obstetrics and Gynecology, Center of Postgraduate Medical Education, cohort). Basic characteristics of the study group are presented in Table 1. All women were Caucasian. There were no significant differences observed between the groups of women in monochorionic and dichorionic pregnancies; 71 women delivered at research centers and information on delivery of the other 8 was collected via a telephone interview.

The values of PlGF, sFlt-1, sFlt-1/PlGF ratio and sEng measured in the first trimester of pregnancy are shown in Table 2. Box-and-whisker plots showing serum biomarkers concentrations are presented in Figure 1. PlGF levels were similar in monochorionic and dichorionic groups. Although serum concentration of sFlt-1 was significantly higher in dichorionic group, no differences in sFlt-1/PlGF ratio were observed. sEng concentrations were similar in both monochorionic and dichorionic groups. No significant differences were observed in serum concentrations of biomarkers depending on the smoking status.

Serum concentrations of PlGF, sFlt-1, sFlt-1/PlGF ratio and sEng in the third trimester of pregnancy are presented in Table 3 and box-and-whisker plots showing biomarkers levels are presented in Figure 2. Similar to the results obtained in the first trimester, no significant differences in PlGF and sEng levels were observed between monochorionic and dichorionic groups. Analogously, sFlt-1 concentration was significantly higher in the dichorionic group. Due to the higher level of sFlt-1, the sFlt-1/PlGF ratio was also significantly higher in the dichorionic group. No significant differences were observed in serum concentrations of biomarkers depending on the smoking status.

In both analyzed groups, significant differences between biomarkers levels in the first and the third trimester of pregnancy were observed. In women in monochorionic gestation the PlGF level was almost 10 times higher in the third trimester of pregnancy in comparison to the first one (*p* < 0.001). sFlt-1 concentration was around two-fold higher in the third trimester (*p* < 0.001). A much bigger increase in PlGF in comparison to sFlt-1 concentration caused significantly lower sFlt-1/PlGF ratio observed in the third trimester in comparison to the first one in the monochorionic group. sEng level in the third trimester was also higher than in the first one (*p* = 0.046). Measurements of biomarker concentration in the first and the third trimester in the monochorionic group are presented in Table 2. No significant relation was observed between gestational weight gain and analyzed biomarkers.

Similar relations were observed in the group of women in dichorionic gestation. PlGF (*p* < 0.001) and sFlt-1 (*p* < 0.001) levels were higher in the third trimester of pregnancy, while the sFlt-1/PlGF ratio was significantly lower in comparison to the first trimester level (*p* < 0.001). sEng concentration was also higher in the third trimester (*p* = 0.001). Measurements of biomarkers concentration in the first and the third trimesters in the dichorionic group are presented in Table 2.

Further analysis after excluding women with gestational hypertension or preeclampsia was conducted (12 cases excluded). The values of PlGF, sFlt-1, sFlt-1/PlGF ratio and sEng measured in the first trimester of pregnancy are shown in Table 3. Similar relations were observed. There were no significant differences in PlGF and sEng levels between monochorionic and dichorionic groups. sFlt-1 concentration was significantly higher in the dichorionic group in both first and third trimesters of pregnancy.

## 4. Discussion

This is the first published study investigating concentrations of serum biomarkers PlGF, sFlt-1, sFlt-1/PlGF ratio and sEng in a cohort of twin gestation in the first and the third trimesters of pregnancy in relation to chorionicity. We found a significantly higher concentration of sFlt-1 in dichorionic in comparison to monochorionic pregnancies in both the first and the third trimesters. PlGF and sEng levels did not differ between mono- and dichorionic gestation in both study periods. sFlt-1 expression was related to twin gestation chorionicity, while PlGF expression was not. PlGF, sFlt-1 and sEng concentrations increased significantly during gestation and were much higher in the third trimester compared to the values measured in the first trimester of pregnancy.

The observed increase in the concentrations of PlGF, sFlt-1 and sEng in the third trimester of pregnancy in comparison to the first trimester stays in line with other published research. Maynard et al. obtained blood samples from women in twin pregnancy between 24 and 36 weeks and found the value of sFlt-1 to be significantly higher in multiples than in singletons and to increase during gestation in multiples. The greatest increase was observed in samples collected at 27–30 weeks and 31–35 weeks of gestation. PlGF concentration was significantly higher in multiple than in singleton gestation and a decline in PlGF concentration between 27–30 weeks and 31–35 weeks was found [20]. In our study we compared biomarker concentrations in only two blood samples collected in the first and the third trimester of pregnancy and no decline in PlGF level was observed. A continuous increase of sFlt-1 plasma concentration from the first to the third trimester of pregnancy was also observed by Ruiz-Sacedón et al. [21]. Faupel-Badger et al. obtained blood samples at the following median weeks of gestation: 9.7, 17.8, 25.9 and 35.1 weeks and found sFlt-1 level to increase with gestational age. A rapid acceleration in sFlt-1 increase was observed beyond 26 weeks of gestation. PlGF concentration in maternal plasma rose from the first trimester to 26 weeks and declined afterwards [22]. It is possible that this decrease of PlGF concentration takes place after 32–34 weeks of gestation and, therefore, it was not noticed in our study.

There are few studies investigating a relationship between angiogenic biomarkers concentrations and chorionicity of a twin gestation and their results are confusing. Until now no research investigating an impact on chorionicity of biomarkers levels over the pregnancy was published. Faupel-Badger et al. measured sFlt-1, PlGF and sEng between 31 and 39 weeks of gestation in a group of 41 women in twin gestation. The authors found maternal concentrations of sFlt-1, sEng and sFlt-1/PlGF ratio to be higher in monochorionic than dichorionic twins after adjustment for gestational age. No significant differences concerning PlGF level were observed [22]. In our study analogous similar concentrations of PlGF between mono- and dichorionics were found at 32 to 34 weeks of twin gestation. Other biomarkers were found to be higher in the dichorionic pregnancy group, which is opposite to the results described by Faupel-Badger et al., however their study group was much smaller. Sánchez et al. performed a prospective study investigating angiogenic factors levels in a group of twin pregnancies and found no significant differences in sFlt-1, PlGF and sEng between monochorionic and dichorionic gestations in the first trimester of pregnancy [9]. However, their study group was also small and included only 12 women in monochorionic gestation. We proved our study group to be large enough according to the power analysis. Svirsky et al. analyzed PlGF levels in maternal serum of women pregnant with twins in the first and second trimesters of pregnancy and found no differences in PlGF concentrations in relation to chorionicity as well [10]. These results are analogous to ours. Cowans and Spencer measured PlGF concentration in 440 dichorionic, 116 monochorionic twin, and 607 singleton pregnancies in the first trimester. They found PlGF levels to be 41% higher in dichorionics, but only 16% higher in monochorionics, compared to singleton pregnancies [23].

Faupel-Badger et al. hypothesized that angiogenic factors release may be related to placental mass. The placental weight is higher in dichorionic than in monochorionic twin pregnancies [22]. The hypothesis presupposes that PlGF and sFlt-1 expression is related to the placental mass. However, Faupel-Badger et al. did not find any correlation between placental weight and sFlt-1 concentration in maternal serum. Such correlation was however described by other authors. Maynard et al. found serum sFlt-1 level to be significantly correlated with total placental weight (R = 0.62, *p* = 0.002), while no such relation between PlGF and placental weight was found (R = 0.26, *p* = 0.15) [20]. A relationship between sFlt-1 and placental weight was also observed by Bdolah et al. [24], while the lack of it by Wathén et al. and Ruiz-Sacedón [21,25]. Faupel-Badger et al. observed a significant correlation between placental mass and PlGF concentration and inverse correlation between placental weight and sFlt-1/PlGF ratio in the third trimester and at delivery in twins [22]. Along with the duration of pregnancy the placental mass increases up until 32–34 weeks of gestation. In our study we found all angiogenic biomarkers to be elevated in the third trimester of pregnancy in comparison to the first one. We assume that the observed increase in PlGF concentration was related to higher placental mass (especially cytotrophoblastic cells amount) in the third trimester. Increase in sFlt-1 serum concentration may also be related to higher placental weight as bigger placenta produces more sFlt-1. However, there may be another explanations. As the placenta grows, a relative reduction of the intervillium space occurs and may lead to trophoblast ischemia and hypoxia. It has been shown that hypoxic syncytiotrophoblast and cytotrophoblast produce an excessive amount of sFlt-1 [26]. As we found a much bigger increase in PlGF than sFlt-1 concentration in the third trimester of a twin pregnancy, we concluded that the observed rise is rather due to higher placental weight rather than placental ischemia or both processes take place. However, if ischemia would be a dominant factor a bigger increase would be expected in sFlt-1 than PlGF concentrations. Therefore, we believe that observed changes in biomarker levels in the third trimester are simply because there is just too much trophoblast.

We found a decreased sFlt-1/PlGF ratio in the third trimester in comparison to the first one in both monochorionic and dichorionic twin pregnancies. In both groups sFlt-1 concentration was about two-fold higher in the third trimester while PlGF levels were about 10 times higher in monochorionic pregnancies and around five times higher in dichorionic pregnancies. Therefore sFlt-1/PlGF ratio was decreased in the third trimester, especially in the monochorionic group.

sFlt-1 was significantly higher in dichorionic gestation in both first and third trimesters of pregnancy. The sFlt-1/PlGF ratio was significantly higher in the dichorionic than in the monochorionic group in the third trimester of the pregnancy. We believe that those findings may help to explain higher incidence of PE observed in dichorionic twin gestation. Higher concentrations of sFlt-1 in maternal serum in the third trimester in twins in comparison to singleton pregnancies were reported by several authors [20,22,24,27,28]. Increased expression of sFlt-1 and increased sFlt-1/PlGF may contribute to PE development in twins, especially dichorionic as the increase is bigger than in monochorionic gestation. This could explain why PE is more frequent in dichorionic twin gestation. Higher concentrations of sFlt-1 in twins in comparison to singletons were reported in the first trimester of pregnancy as well [9,22]. According to Faupel-Badger et al., this may be due to more shallow placentation while twinning [22]. We observed higher sFlt-1 levels in dichorionic than monochorionic twin pregnancies in the first trimester as well. On the basis of these findings, we hypothesize that more frequent PE development in dichorionics may have a complex etiology: as sFlt-1 is higher in the first and the third trimesters, the time during which Sflt-1 affects maternal vessels may play a crucial role in PE development.

The strength of our study is its uniqueness as it is the first one investigating concentrations of angiogenic biomarkers in a twin pregnancy in the first and the third trimesters of pregnancy in relation to chorionicity. According to the study methods, both collections of blood samples were made in the same group of patients in both gestational periods and from all women included in the analysis. Women were counselled according to one uniform policy and pregnancy complications were well defined. However, our study has limitations. It is a two-center investigation. Data on placental weight were unavailable. There were only three cases of PE in our study group which made it impossible to investigate the relation between angiogenic factors and PE occurrence. All the participants were Caucasian, which limits the external validity of this study. Only two samples of blood were collected during the first and the third trimesters of pregnancy, therefore no data on the second trimester are available.

## 5. Conclusions

In conclusion, we have shown that angiogenic biomarker levels change during twin pregnancy and differ between dichorionic and monochorionic twin gestation. The concentration of sFlt-1 is related to the chorionicity of a twin pregnancy. Further large prospective studies are needed to establish if the observed higher concentration of sFlt-1 and sFlt-1/PlGF ratio in dichorionic pregnancy can contribute to PE development in twin gestation.

## Figures and Tables

**Figure 1 diagnostics-11-01181-f001:**
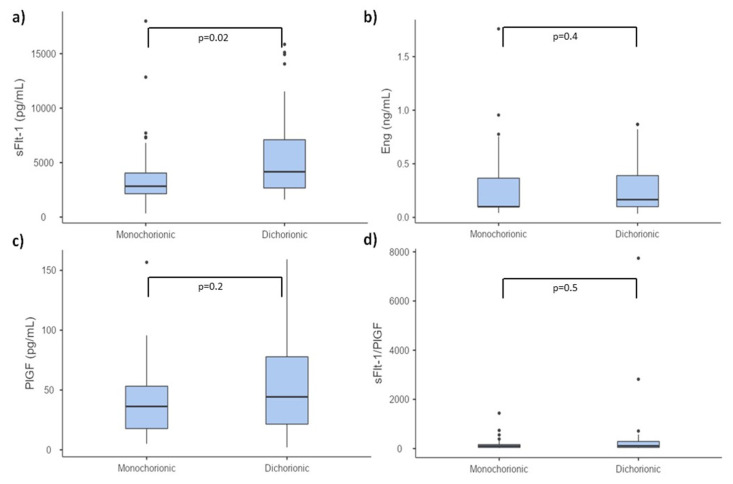
Box-and-whisker plots showing serum values of PlGF, sFlt-1, sFlt-1/PlGF ratio and sEng measured in the first trimester of twin pregnancy. (**a**) sFlt-1; (**b**) sEng; (**c**) PlGF; (**d**) sFlt-1/PlGF serum concentration.

**Figure 2 diagnostics-11-01181-f002:**
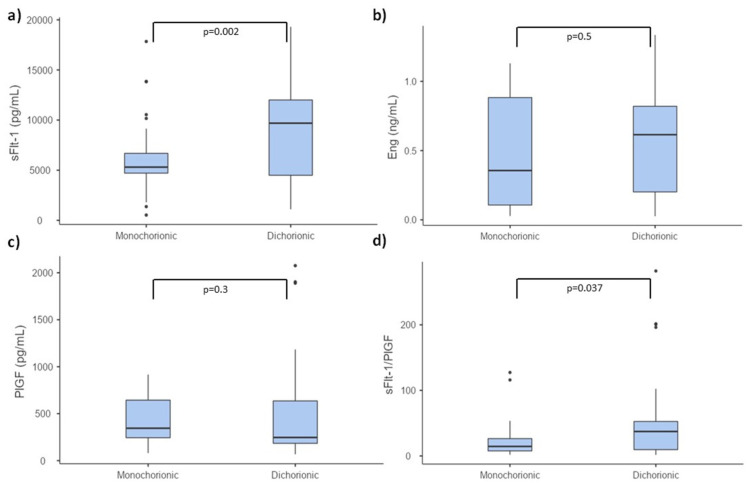
Box-and-whisker plots showing serum values of PlGF, sFlt-1, sFlt-1/PlGF ratio and sEng measured in the third trimester of twin pregnancy. (**a**) sFlt-1; (**b**) sEng; (**c**) PlGF; (**d**) sFlt-1/PlGF serum concentration.

**Table 1 diagnostics-11-01181-t001:** Basic characteristics of the study group.

	Study Group N = 79	Monochorionic Pregnancies N = 43	Dichorionic Pregnancies N = 36	*p*
	N (%)	N (%)	N (%)	
Age (years) *	31.5 ± 4.3	31.8 ± 4.3	31.3 ± 4.3	0.7
Primiparity	39 (49.4)	23 (53.5)	16 (44.4)	0.3
Smoking status				0.8
current	3 (3.8)	2 (4.7)	1 (2.8)	
past	45 (57)	23 (53.4)	22 (61.1)	
never	28 (35.4)	16 (37.2)	12 (33.3)	
unknown	3 (3.8)	2 (4.7)	1 (2.8)	
Pre-gravid BMI (kg/m^2^) *	23.8 ± 4.6	23.1 ± 4.6	24.6 ± 4.6	0.08
Obesity	8 (10.1)	4 (9.3)	4 (11.1)	0.4
Gestational weight gain *	18.7 ± 8.6	17.8 ± 8.9	19.6 ± 8.4	0.3
ART	17 (21.5)	7 (16.3)	10 (27.8)	0.3
Gestational hypertension	9 (11.4)	4 (9.3)	5 (13.9)	0.7
Early PE	2 (2.5)	0	2 (5.6)	0.3
Late PE	1 (1.3)	0	1 (2.8)	0.8
GDM	21 (26.6)	11 (25.6)	10 (27.8)	0.5
Gestational age at delivery (weeks) *	35.7 ± 1.9	35.5 ± 1.7	35.9 ± 2.2	0.4
Cesarean delivery	46 (58.2)	24 (55.8)	22 (61.1)	0.8
First twin birthweight (g) *	2423 ± 465	2383 ± 488	2472 ± 441	0.7
First twin 5th minute Apgar ≤ 7 points	3 (3.8)	2 (4.6)	1 (2.8)	0.6
Second twin birthweight (g) *	2360 ± 409	2314 ± 420	2417 ± 397	0.3
Second twin 5th minute Apgar ≤ 7 points	6 (7.6)	3 (7)	3 (8.3)	0.9
Discordant inter-twin birthweight	7 (8.9)	4 (9.3)	3 (8.3)	0.4

*—average ± standard deviation; BMI—body mass index; ART—assisted reproductive technology; PE—preeclampsia; GDM—gestational diabetes mellitus.

**Table 2 diagnostics-11-01181-t002:** Serum concentrations of placental growth factor (PlGF), serum soluble fms-like tyrosine kinase-1 (sFlt-1), sFlt-1/PlGF and endoglin (Eng) measured in the first trimester and the third of pregnancy in monochorionic and dichorionic groups.

	1st Trimester	3rd Trimester	*p*	1st Trimester	3rd Trimester	*p*
	Median (Interquartile Range)	Median (Interquartile Range)		Median (Interquartile Range)	Median (Interquartile Range)	
	Monochorionic			Dichorionic		
PlGF (pg/mL)	36.3 (17.8–53.2)	345 (243–644)	<0.001	44.3 (21.5–77.8)	246.7 (184–635)	<0.001
sFlt-1 (pg/mL)	2830.9 (2140–4042)	5306 (4706–6679)	<0.001	4156.4 (2668–7099)	9692 (4489–12004)	<0.001
sFlt-1/PlGF	99.1 (45.1–160.1)	14.5 (7.7–26.5)	<0.001	108.1 (45.3–288.2)	37.3 (9.7–52.6)	<0.001
Eng (ng/mL)	0.1 (0.1–0.4)	0.36 (0.1–0.9)	0.046	0.17 (0.1–0.4)	0.62 (0.2–0.8)	0.001

**Table 3 diagnostics-11-01181-t003:** Serum concentrations of PlGF, sFlt-1, sFlt-1/PlGF and Eng measured in the first trimester and the third of pregnancy in monochorionic and dichorionic groups without hypertension-related disorders.

	1st Trimester	3rd Trimester	*p*	1st Trimester	3rd Trimester	*p*
	Median (Interquartile Range)	Median (Interquartile Range)		Median (Interquartile Range)	Median (Interquartile Range)	
	Monochorionic			Dichorionic		
PlGF (pg/mL)	36.5 (17.8–54.1)	348 (245–652)	<0.001	48.7 (23.5–79.1)	257 (184–644)	<0.001
sFlt-1 (pg/mL)	2829.2 (2140–4041)	5300 (4703–6671)	<0.001	4112.1 (2651–7083)	9653 (4411–11752)	<0.001
sFlt-1/PlGF	98.2 (45–159.1)	14.4 (7.5–26.5)	<0.001	103.9 (42.8–265.7)	35.9 (9.2–49.6)	<0.001
Eng (ng/mL)	0.1 (0.1–0.4)	0.35 (0.1–0.9)	0.049	0.14 (0.1–0.4)	0.6 (0.2–0.9)	0.001

## Data Availability

The data presented in this study are available on request from the corresponding author.

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
