# Peer review of "sFlt-1, Not PlGF, Is Related to Twin Gestation Choronicity in the First and Third Trimesters of Pregnancy"

_diagnostics, 2021, doi:10.3390/diagnostics11071181_

Round 1

Reviewer 1 Report

I suggest you make Table 2 clearer. Maybe you would do one for the first trimester and one for the third trimester.

Express the statistical significance "p" with the capital letter P.

Can you also use your data to relate to the proportion of weight gain during pregnancy?

You have specified a smoker status. Do you have data on ex-smokers how much time has passed since quitting? Or you can calculate the difference between the values of smokers and ex-smokers versus those who have never smoked. 

Author Response

Honourable Reviewer,

Thank you for your time and all the valuable suggestions which helped us improve the paper. Here are the answers to your comments:

  1. I suggest you make Table 2 clearer. Maybe you would do one for the first trimester and one for the third trimester.

When we divided Table 2 into two separate tables including results concerning the first and the third trimester of pregnancy, we could present the same data that were visualized in figures 1 and 2, while it was difficult to present comparison between monochorionic and dichorionic pregnancies in each trimester. We believe it was even more complex and confusing this way, so we decided not to make two separate tables.

  1. Express the statistical significance "p" with the capital letter P.

It has been modified according to the Reviewer’s suggestion.

  1. Can you also use your data to relate to the proportion of weight gain during pregnancy?

No significant relation was observed between gestational weight gain and analyzed biomarkers. This information was added to the manuscript.

  1. You have specified a smoker status. Do you have data on ex-smokers how much time has passed since quitting? Or you can calculate the difference between the values of smokers and ex-smokers versus those who have never smoked.

Smoking status was declared by participants. Past smoking habit was defined as quitting smoking at least one month before the current pregnancy. We only used the cut-off of one month period before the pregnancy without more specific information on how long before the pregnancy women stopped smoking. Therefore, we do not have exact data on ex-smokers. No significant differences were observed in serum concentrations of biomarkers depending on the smoking status. It was added to the manuscript.

Reviewer 2 Report

The manuscript submitted by Kozlowski and colleagues aims to fill a gap in the research literature by ascertaining the gestational expression of PlGF, sFlt-1 and sEng in both mono- and dichorionic twin pregnancies. As has been done by other groups many times before, the authors collected maternal blood during gestation and ran a few ELISAs. The current manuscript provides only slight variation on that playbook, by recruiting mothers carrying twins. The authors find that all factors increase between the first and third trimesters, and that only sFlt-1 is higher in dichorionic vs. monochorionic twin pregnancies. However, this higher expression of sFlt-1 is not related to any known pathology, and all other parameters measured were equivalent between groups. As such this manuscript provides little new information. Overall, there are several limitations to the paper that should be addressed.

1) The authors speculate that the increased expression of the 3 factors may be related to higher placental mass in dichorionic vs monochorionic pregnancies. However, there is no data on the placentas at all. Did the authors collect any placental samples for histology that could be analyzed? Is there any placental measurements that might be gleaned from ultrasound scans?

2) There is very little information on fetal development during these pregnancies, nor is there much information about the infants, other than birth weight. Considering that there were no significant differences in GHTN, PE, or GDM between the groups, there might be some sub-clinical findings that may exist. For instance, CRL measurements, or other fetal development parameters could be included. Also, Apgar scores could be included.

3) BMI was collected during the study, but it is not clear when. Also, were these self-reported or measured directly?

4) Same issue for gestational weight gain, was pre-gravid weight self-reported or measured?

5) All study participants were Caucasian, which severely limits the external validity of this study.

6) Only 2 blood samples were taken from each participant. Many similar studies take 4 or more blood samples during gestation. This really limits the scope of this paper.

7) The authors comment that placental hypoxia may cause increased expression of sFlt-1 but provide no evidence that their study participants exhibited placental hypoxia.

Author Response

Honourable Reviewer,

Thank you for your time and all the valuable suggestions which helped us improve the paper. Here are the answers to your comments:

  • The authors speculate that the increased expression of the 3 factors may be related to higher placental mass in dichorionic vs monochorionic pregnancies. However, there is no data on the placentas at all. Did the authors collect any placental samples for histology that could be analyzed? Is there any placental measurements that might be gleaned from ultrasound scans?

Unfortunately, no data on placental mass, samples for placental histology or ultrasound measurements of all women for the study group is available. Therefore, we can only refer to other published research, which proves that the placental weight is higher in dichorionic than in monochorionic twin pregnancies. It was referred in the manuscript.

  • There is very little information on fetal development during these pregnancies, nor is there much information about the infants, other than birth weight. Considering that there were no significant differences in GHTN, PE, or GDM between the groups, there might be some sub-clinical findings that may exist. For instance, CRL measurements, or other fetal development parameters could be included. Also, Apgar scores could be included.

It was added to the Table 1.

  • BMI was collected during the study, but it is not clear when. Also, were these self-reported or measured directly?

4) Same issue for gestational weight gain, was pre-gravid weight self-reported or measured?

BMI was calculated by dividing the pre-gravid body mass by the square of the body height. Pre-gravid weight was self-reported, while weight at delivery was measured by the hospital stuff. This information was added to the manuscript.

  • All study participants were Caucasian, which severely limits the external validity of this study.

It was added to the limitations of the study.

  • Only 2 blood samples were taken from each participant. Many similar studies take 4 or more blood samples during gestation. This really limits the scope of this paper.

2 samples of blood were collected from all the participants, one in the first and the second one in the third trimester of pregnancy. All the collected blood was used for analysis presented in the study. No samples were available in the second trimester. This limitation was added to the manuscript.

  • The authors comment that placental hypoxia may cause increased expression of sFlt-1 but provide no evidence that their study participants exhibited placental hypoxia.

According to published research on sFlt-1 in singleton pregnancy it has been proven that sFlt-1 is a marker on trophoblast hypoxia. However not every pregnancy with elevated sFlt-1 is complicated by clinical signs of placental hypoxia like preeclampsia or fetal growth restriction. There were only 3 cases of preeclampsia in our study group, all in dichorionic pregnancies.